# Dynamic changes of the Prf/Pto tomato resistance complex following effector recognition

Arsheed H. Sheikh[1,2,10], Iosif Zacharia[1,10], Alonso J. Pardal[1],
Ana Dominguez-Ferreras[1], Daniela J. Sueldo[1,3], Jung-Gun Kim[4],
Alexi Balmuth[5,6], Jose R. Gutierrez[6], Brendon F. Conlan[7], Najeeb Ullah[1],
Olivia M. Nippe[1], Anil M. Girija[8], Chih-Hang Wu[6], Guido Sessa[8],
Alexandra M. E. Jones[1], Murray R. Grant[1], Miriam L. Gifford[1,9],
Mary Beth Mudgett[4], John P. Rathjen[7] & Vardis Ntoukakis[1,9] ✉

In both plants and animals, nucleotide-binding leucine-rich repeat (NLR) immune receptors play critical roles in pathogen recognition and activation of innate immunity. In plants, NLRs recognise pathogen-derived effector proteins and initiate effector-triggered immunity (ETI). However, the molecular mechanisms that link NLR-mediated effector recognition and downstream signalling are not fully understood. By exploiting the well-characterised tomato Prf/Pto NLR resistance complex, we identified the 14-3-3 proteins TFT1 and TFT3 as interacting partners of both the NLR complex and the protein kinase MAPKKKα. Moreover, we identified the helper NRC proteins (NLR-required for cell death) as integral components of the Prf/Pto NLR recognition complex. Notably our studies revealed that TFTs and NRCs interact with distinct modules of the NLR complex and, following effector recognition, dissociate facilitating downstream signalling. Thus, our data provide a mechanistic link between activation of immune receptors and initiation of downstream signalling cascades.

Plants have evolved two major strategies to mount an immune response against pathogenic bacteria[1]. The first level of immunity is activated by conserved microbial factors called MAMPs (microbe-associated molecular patterns), recognized by membrane-localised pattern recognition receptors (PRRs). Following recognition, MAMP-triggered immunity (MTI) is activated and communicated by an array of downstream signalling responses including production of reactive oxygen species (ROS), activation of mitogen-activated protein kinases (MAPKs), chromatin remodelling and activation of defence genes[2,3]. To overcome MTI, pathogens translocate a repertoire of virulence proteins called effectors to their hosts to overcome immunity by targeting various signalling modules. To counteract the action of effectors, plants have evolved cytosolic resistance (R) proteins capable of recognising effectors and generating a defence referred to as effector-triggered immunity (ETI)[1]. Intracellular R proteins typically contain a central nucleotide-binding (NB) domain and a C-terminal leucine rich repeats (LRR) domain but have variable N-terminal domains (NLRs). ETI is very robust and usually leads to strong and prolonged activation

[1]School of Life Sciences, University of Warwick, Coventry CV4 7AL, UK. [2]Center for Desert Agriculture, BESE Division, King Abdullah University of Science and Technology, Thuwal 23955-6900, Saudi Arabia. [3]Department of Biology, Faculty of Natural Sciences, Norwegian University of Science and Technology, Hogskoleringen 1, 7491 Trondheim, Norway. [4]Department of Biology, Stanford University, Stanford, CA 94305, USA. [5]J.R. Simplot Company, Boise, ID, USA. [6]The Sainsbury Laboratory, Norwich Research Park, Norwich NR4 7UH, UK. [7]Research School of Biology, The Australian National University, Acton 2601 ACT, Australia. [8]School of Plant Sciences and Food Security, Tel-Aviv University, 69978 Tel-Aviv, Israel. [9]Warwick Integrative Synthetic Biology Centre, University of Warwick, Coventry CV4 7AL, UK. [10]These authors contributed equally: Arsheed H. Sheikh, Iosif Zacharia. ✉e-mail: v.ntoukakis@warwick.ac.uk

of MAPKs[4], culminating in the activation of programmed cell death (PCD) which restricts the growth of pathogens[5]. Recent advances have demonstrated that NLR activation induces oligomerisation and the formation of high molecular weight complexes, the resistosomes[6–8]. An additional level of ETI complexity has been proposed wherein a group of helper NLR proteins are not directly involved in effector recognition but rather have evolved to mediate signalling from multiple sensor NLRs[9–11].

In tomato (*Solanum lycopersicum*), the NLR receptor complex Prf/Pto[12] recognises the *Pseudomonas syringae* pv. *tomato* (*Pst* DC3000) bacterial effectors AvrPto and AvrPtoB that target multiple PRRs[13,14]. Prf is a NLR protein with an unusual N-terminal domain, a Solanaceae (SD) domain, a coiled-coil (CC) domain and a classical NB-LRR domain. Pto is a serine/threonine kinase which directly binds AvrPto and AvrPtoB and initiates downstream signalling through a partially understood mechanism[12,15]. The N-terminal domain of Prf serves as a platform for Prf oligomerisation and interaction with Pto[16] bringing two molecules of Pto into close proximity and holding them in an inactive conformation in the absence of the effectors[17]. Effector binding by the sensor Pto removes the Prf-imposed negative regulation and causes disruption of the Pto kinase P + 1 loop[18]. This de-repression activates a second Pto kinase within the same Prf/Pto complex, the helper, which then transphosphorylates the sensor kinase at T199, thereby fully activating the complex[17]. Activation of the Prf/Pto complex leads to activation of MAPK signalling cascades[19]. Two MAPKKKs, MAPKKKα and MAPKKKε, are activated following effector recognition by the Prf/Pto complex[20,21]. Furthermore, two distinct MAPKs, MPK2 and MPK3, which are tomato orthologs of SIPK and WIPK, are reported to be activated by MAPKKKα in *N. benthamiana*[22]. One possible mechanism by which MAPKKK activation might be regulated following activation of the Prf/Pto complex was attributed to the function of 14-3-3 proteins[23].

14-3-3 proteins are conserved eukaryote-specific proteins which act as phosphorylation sensors. They bind to their target proteins by sensing their phosphorylation status and modulate their function. 14-3-3 proteins can regulate either the dynamics of protein-protein interactions or the subcellular localization of target proteins[24]. They can also act as scaffolds to bridge two proteins[25]. The tomato 14-3-3 (TFT) isoform TFT7 positively regulates PCD mediated by multiple NLRs including Prf. TFT7 interacts with the C-terminus of MAPKKKα to induce PCD, most likely by regulating its stability[19]. TFT7 also interacts with the downstream MKK2, to positively regulate PCD[26]. Despite the proposed role of MAPKs and TFTs in Prf/Pto- mediated signalling, it is not known how effector recognition from the NLR complex is transduced to MAPKKK activation. Furthermore, two helper NLRs called NRCs (NLR-required for cell death proteins), NRC2 and NRC3, were also found to be necessary for cell death following activation of the Prf/Pto complex[9,27] and were shown to form resistosomes[28]. However, is not yet clear if these helper NRCs directly interact with Prf or other components of the Prf/Pto complex.

Here we show that TFT proteins and NRCs interact with distinct components of the Prf/Pto complex. We provide evidence that TFT proteins interact with Pto and MAPKKKα while NRCs interact with Prf. Moreover, we show that following AvrPto/AvrPtoB effector recognition the Prf/Pto complex dissociates into distinct modules. The Prf-Pto and TFT-MAPKKKα interactions remain intact while the Prf-Prf, Prf-NRCs and Pto-TFT interactions dissociate. We propose a model where dissociation of MAPKKKα from the Prf/Pto complex relieves the negative repression of MAPKKKα, allowing activation of the downstream MAPK pathway, while at the same time NRCs dissociate from Prf potentially enabling resistosome formation.

## Results

### Tomato 14-3-3 proteins interact with Pto and downstream MAPKKKs

The tomato 14-3-3 protein TFT7 was previously shown to interact with MAPKs and positively regulate Pto-mediated PCD[19,26]. In order to determine whether TFT proteins physically interact with components of the Prf/Pto complex, we performed a series of co-immunoprecipitation and split-luciferase experiments with Pto or Prf and the tomato TFT isoforms (TFT1 to TFT11) in *N. benthamiana*. We found that Pto preferentially interacted with TFT1 and TFT3 (Fig. 1a and Supplementary Fig. 1a). By contrast, Prf did not interact with TFT1 or TFT3 (Supplementary Fig. 1b). Further, co-immunoprecipitations showed that Pto interaction with TFT1 was diminished once TFT3 was co-expressed (Fig. 1b), suggesting that there is intrinsic competition between TFT1 and TFT3 for Pto binding. This preferential binding of Pto to TFT3 is dependent on the kinase activity of Pto, since the kinase dead Pto[D164N] interact with TFT1 (Supplementary Fig. 2a) but showed no apparent preference for TFT1 or TFT3 (Fig. 1b). However,

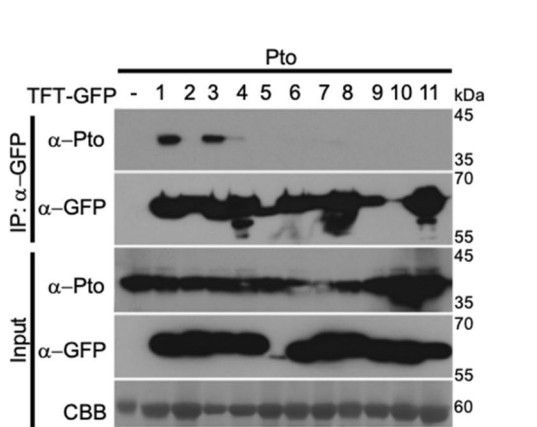

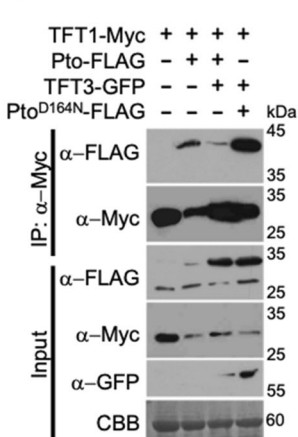

**Fig. 1 | Pto kinase interacts with the tomato 14-3-3 proteins TFT1 and TFT3.**
**a** Interaction of Pto with tomato 14-3-3s. Pto was co-expressed with 14-3-3 proteins from tomato (TFTs). TFT(1-11)-GFP and Pto constructs were transiently expressed in *N. benthamiana* leaves and TFTs were immunoprecipitated (IP) using GFP-Trap Agarose beads. **b** Impact of TFT3 on Pto-TFT1 complex formation. TFT1-Myc, Pto-FLAG, TFT3-GFP and Pto[D164N]-FLAG (a kinase-dead variant of Pto) constructs were transiently expressed in *N. benthamiana* leaves. TFT1-Myc was immunoprecipitated with Myc-Trap agarose beads. All leaves were harvested 3 days post infiltration for protein extraction and immunoblots were performed with the antibodies indicated on the left of each panel. Coomassie Brilliant Blue (CBB) staining of the membranes (lowest row on each panel) was used as loading control. The experiments were repeated three times and typical results are shown.

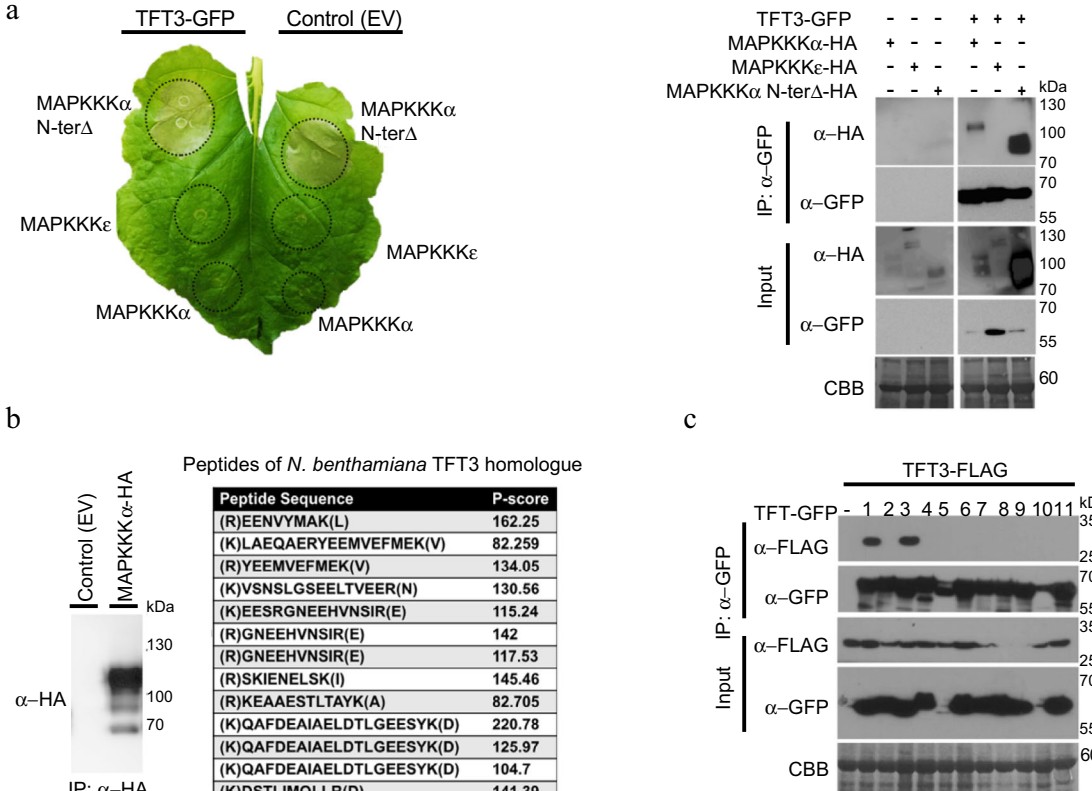

**Fig. 2 | TFT3 interacts with MAPKKKα and homo and hetero-dimerises with TFT1. a** TFT3 interacts with tomato MAPKKKα but not tomato MAPKKKε. TFT3-GFP or control Empty Vector (EV) were co-expressed with HA tagged MAPKKKα, MAPKKKε or the autoactive N-terminal deleted version of MAPKKKα in *N. benthamiana* leaves. Immunoprecipitation (IP) of TFT3 was carried out using GFP-Trap agarose beads and the corresponding interactions were identified by immunoblotting. **b** Tomato MAPKKKα interacts with the *N. benthamiana* homologue of TFT3. MAPKKKα-HA was transiently expressed in *N. benthamiana* leaves and immunoprecipitated using α-HA agarose beads. 9 unique peptides of *N.* *benthamiana* TFT3 homologue were identified by liquid chromatography–mass spectrometry. **c** TFT3 forms homo and hetero-dimers with TFT1. TFT3-FLAG was transiently co-expressed with TFT(1-11)-GFP in *N. benthamiana* leaves. Immunoprecipitation of TFTs was carried out using GFP-Trap agarose beads and the interaction with TFT3 was identified by immunoblotting. All leaves were harvested 3 days post-infiltration for protein extraction and immunoblots were performed with the antibodies indicated on the left. Coomassie Brilliant Blue (CBB) staining of the membranes verified equal protein loading. The experiments were repeated three times and typical results are shown.

the kinase activity of Pto is not a prerequisite for binding since Pto[D164N], the kinase dead but constitutively active Pto mutant (Pto[L205D]) and the transphosphorylation deficient Pto mutant (Pto[S198T199/AA])[17] could each interact with TFT3 (Supplementary Fig. 2b).

We next investigated the interaction of TFT3 with two well-characterized tomato MAPKKKs MAPKKKα and MAPKKKε, which are important for PCD initiation following activation of the Prf/Pto complex[20,21]. Co-immunoprecipitation experiments showed that TFT3 interacted with MAPKKKα and the N-terminal-deleted autoactive version of MAPKKKα (Supplementary Fig. 3), but not with MAPKKKε (Fig. 2a). The interaction was further confirmed by performing split-luciferase experiments (Supplementary Fig. 4a) and mass spectrometry analysis of tomato MAPKKKα interacting proteins in *N. benthamiana*. One of the identified interacting proteins of MAPKKKα was the *N. benthamiana* homologue of TFT3 (Fig. 2b), reflecting the conserved nature of these interactions within the *Solanaceae* family. However, no interaction was observed between MAPKKKs and Pto (Supplementary Fig. 4b). Given that 14-3-3 proteins are known to function as both homo and hetero-dimers[23], we hypothesized that either a TFT3 homo-dimer or a TFT1-TFT3 hetero-dimer could possibly act as a bridge between Pto and MAPKKKα. We found that TFT3 was able to form homo-dimers and hetero-dimers with TFT1 (Supplementary Fig. 5), but did not interact with any other TFT isoforms (Fig. 2c). The interaction of TFT1 with other TFTs was less specific as it interacted and formed heterodimers with all tested TFTs (Supplementary Fig. 6).

The protein-protein interactions between Pto-TFT3 and TFT3-MAPKKKα raised the possibility that TFT3 or MAPKKKα could be phosphorylated by Pto. To test this hypothesis, we performed in vitro kinase assays using recombinant His tagged MAPKKKα, Pto, TFT1 and TFT3; all partially purified from *E.coli*. As expected, we observed strong autophosphorylation of MAPKKKα (Supplementary Fig. 7a). Interestingly, Pto and both TFT1 and TFT3 were phosphorylated by MAPKKKα (Supplementary Fig. 7a). However, we did not observe direct phosphorylation of TFT1 or TFT3 by Pto (Supplementary Fig. 7b).

Taken together, our protein-protein interaction data support a model in which TFT3 and TFT1 interact with the Pto component of the Prf/Pto complex and MAPKKKα. Based on this model, we hypothesise that following effector recognition, activation of the Prf/Pto complex leads to TFT-mediated MAPKKKα de-repression and autophosphorylation, rather than activation of MAPKKKα by direct Pto-mediated phosphorylation.

## 14-3-3 proteins regulate tomato ETI responses

The observed interaction of 14-3-3 proteins with components of the NLR complex in tomato and their known role in controlling MAPK activation suggested that they play a central role in Prf/Pto-mediated ETI. To further understand the role of 14-3-3 proteins in ETI, we investigated the immunity phenotypes of 14-3-3 tomato mutants. We first generated CRISPR/Cas9 tomato *tft3* mutant lines in tomato Rio Grande 76R-*prf3* (*prf3*) lines complemented with a *Prf* transgene fused

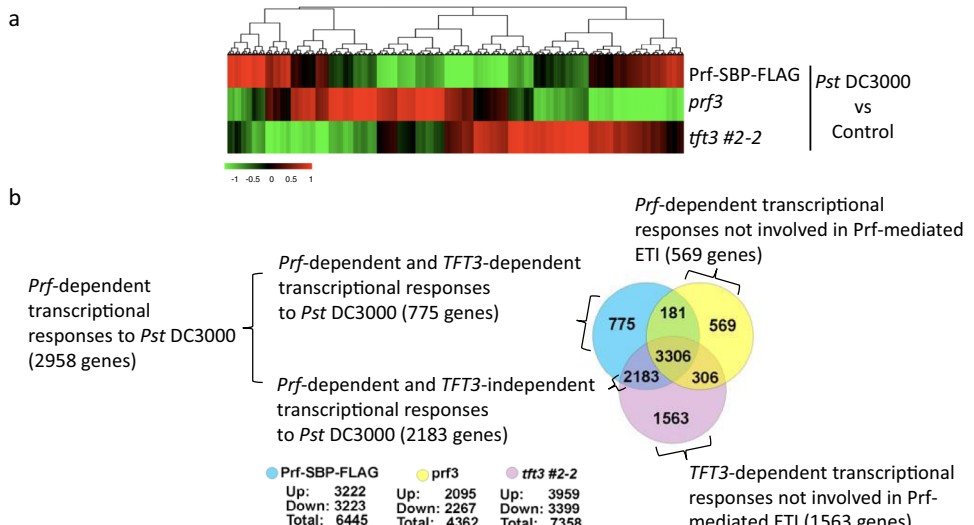

**Fig. 3 | TFT3 contributes to Prf/Pto-mediated transcriptional immune responses. a** ETI transcriptional responses are compromised in tomato *tft3* #2-2 and *prf3* mutants. Heat map showing the expression profile of wild type and mutant tomato lines 6 h post infiltration with *Pseudomonas syringae* pv. *tomato* (*Pst* DC3000). Rio Grande 76 R tomato lines, *prf3*, *prf3*/Prf-SBP-FLAG (Prf-SBP-FLAG) and *prf3*/*tft3 2-2*/Prf-SBP-FLAG (*tft3* #2-2) were either mock inoculated (control) or infiltrated with *Pst* DC3000. Microarray hybridization on custom-designed Agilent microarrays was conducted 6 h post infiltration. Gene expression changes are coloured depending on whether mRNA abundance is relatively increased (red) or decreased (green) within infected leaves compared to mock-inoculated leaves. **b** TFT3 regulates the expression of a subset of Prf-mediated transcriptional responses. Venn diagram shows the numbers of differentially expressed genes (DEGs) and the direction of differential expression in each genotype 6 h after *Pst* DC3000 infiltration in comparison to mock-inoculated conditions. Threshold for was set at greater than 2-fold and transcripts were termed to be DEGs if they showed a Benjamini-Hochberg adjusted *P*-value ≤ 0.05 in the comparison between treatment and control.

to a nucleotide sequence encoding tandem streptavidin binding peptide (SBP) and FLAG epitope tags (*prf3*/*Prf-SBP-FLAG*)[18]. The tomato *prf3* genotype has a 1 kb deletion in the *Prf* gene, rendering it unable to mediate activation of ETI, and the *Prf-SBP-FLAG* transgene fully complements the *prf3* mutation[18]. Two *tft3* mutant lines were generated and only the *tft3* #2-2 line (*tft3 2-2*/*prf3*/*Prf-SBP-FLAG*) was impaired for *TFT3* mRNA expression (Supplementary Fig. 8a). The *tft3* #2-2 mutant has a 250 bp deletion in the coding region of the gene (Supplementary Fig. 8b) and the loss of *TFT3* expression (Supplementary Fig. 8a) suggests that it is a null mutant.

To assess the immunity phenotypes of the generated lines, we performed transcriptomic experiments, MAPK activation assays, photosystem II (PSII) activity measurements, electrolyte leakage and bacterial growth assays. We first performed comparative transcriptomics between *prf3*/*Prf-SBP-FLAG* (containing wild type TFT3) and *tft3* #2-2 line treated with either buffer (as control) or *Pst* DC3000 (Fig. 3a). *Pst* DC3000 expresses both *AvrPto* and *AvrPtoB* effectors and is avirulent on the *prf3*/*Prf-SBP-FLAG* line due to effector recognition by the Prf/Pto complex. We used the susceptible *prf3* tomato line as a control for our infections. Following *Pst* DC3000 inoculation, we identified 6445 Differentially Expressed Genes (DEGs, adjusted *P*-value < 0.05, fold-change > 2) in response to *Pst* DC3000 infection in the *prf3*/*Prf-SBP-FLAG* resistant line compared to buffer control. From these 6445 DEGs, 3487 were also DEGs in the susceptible *prf3* mutant line, therefore their expression did not rely on the presence of the *Prf* gene. In contrast, 2958 genes that were DEGs in the *prf3*/*Prf-SBP-FLAG* line were not differentially expressed in the *prf3* mutant line, hence these genes represent the *Prf*-dependent transcriptional responses to Pst DC3000 (Fig. 3b). The majority of these genes (2183 of the 2958, 74%) were also DEGs in the *tft3* #2-2 mutant line, hence their expression is *Prf*-dependent but *TFT3*-independent. In contrast, 775 (26%) genes were differentially expressed only if both *Prf* and *TFT3* genes were present, hence their expression is *Prf* and *TFT3*-dependent (Fig. 3b). Further inspection revealed that the set of the *Prf* and *TFT3*-dependent genes was enriched for Gene Ontology (GO) terms associated with

catabolic processes (Fisher Exact test, FDR < 0.05), suggesting that TFT3 may regulate catabolic pathways involved in programmed cell death (Supplementary Fig. 9 and Supplementary Data 1). Similar analyses for the *Prf*-dependent and *TFT3*-independent genes indicated GO terms associated with secondary metabolism (Supplementary Fig. 9). Our transcriptomic analysis also revealed that in the absence of *Prf* or *TFT3*, a distinct set of genes is differentially regulated most likely due to the virulence function of the *Pst* DC3000 effectors and the role of TFT3 in other biological processes (Fig. 3b; Supplementary Fig. 9 and Supplementary Data 1). Nevertheless, the 569 and 1563 DEGs in the *prf3* and *tft3* #2-2 mutant lines (Fig. 3b) are not involved in *Prf*-mediated ETI responses since they are not part of the transcriptomic responses of the control *prf3*/*Prf-SBP-FLAG* lines to *Pst* DC3000. Overall, our transcriptomic analysis revealed a bifurcation of the *Prf*-dependent transcriptional responses to *Pst* DC3000 with one branch *TFT3*-dependent and the other *TFT3*-independent (Fig. 3).

Consistent with our data showing that TFT3 positively regulates Prf/Pto-mediated ETI signalling (Fig. 3) through its interaction with MAPKKα (Fig. 2), infection with *Pst* DC3000 resulted in reduced activation of MAPKs (MPK3 and MPK6) in the *tft3* #2-2 mutant line compared to the *tft3* #1-4 (*tft3 1-4*/*prf3*/*Prf-SBP-FLAG*) line expressing *TFT3* (Supplementary Fig. 8a) and the *prf3*/*Prf-SBP-FLAG* line (Fig. 4a). Given the critical role of MPK3/MPK6 in regulating ETI robustness by inhibiting plant photosynthesis[29], we next measured $F_v/F_m$, the maximum dark-adapted quantum efficiency of photosystem II (PSII) a non-destructive measurement related to photosynthetic activity that is modified by *Pst* DC3000 effectors and ETI[30,31]. Following *Pst* DC3000 infection, the attenuated MAPK activation in the *prf3* and *tft3* #2-2 mutant lines (Fig. 4a) resulted in reduced suppression of PSII compared to the *prf3*/*Prf-SBP-FLAG* control line (Fig. 4b). To further investigate the role of TFT3 in Prf/Pto-mediated tomato immunity, we performed electrolyte leakage as a proxy of PCD formation and bacterial growth assays. We observed a small but significant reduction of electrolyte leakage for *tft3* #2-2 mutant plants starting at 6 h post inoculations with *Pst* DC3000 in comparison with *prf3*/*Prf-SBP-FLAG*

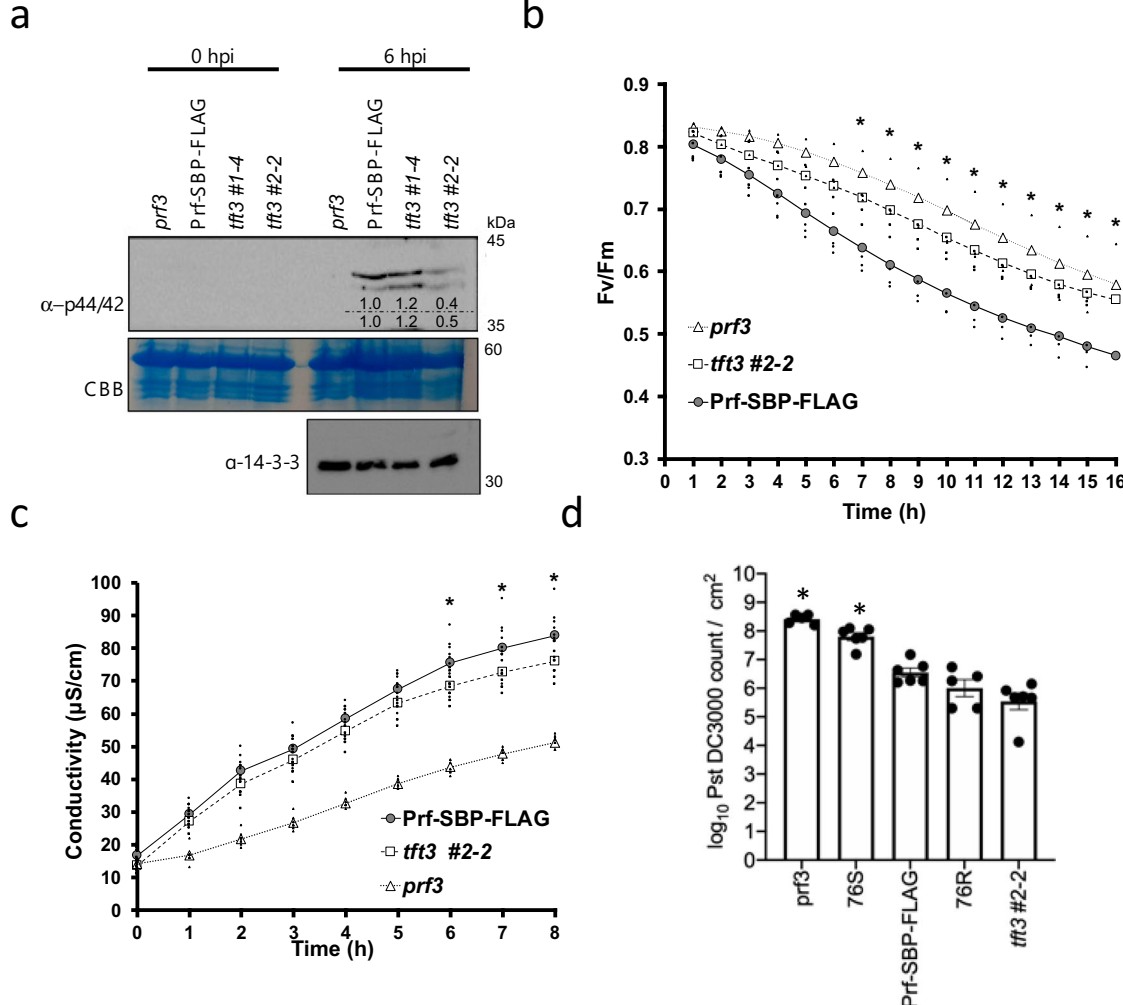

**Fig. 4 | TFT3 contributes to Prf/Pto-mediated ETI. a** MAPK activation is reduced in tomato *tft3 2-2* mutants. Tomato leaves were infiltrated with *Pseudomonas syringae* pv. *tomato* (*Pst* DC3000) and MAPK phosphorylation was detected using an α -phospho-p44/42 antibody at the indicated times. *prf3/tft3 2-2*/Prf-SBP-FLAG (*tft3#2-2*) is a *tft3* mutant but *prf3/tft3 1-4*/Prf-SBP-FLAG (*tft3#1-4*) is not a *tft3* mutant and serves as a transgenic control line. Coomassie Brilliant Blue (CBB) staining of the membrane and an α -14−3−3 immunoblot were used to verify equal protein loading. The numbers above the gel lanes represent the relative phosphorylated protein level, which was determined from the band intensity using ImageJ and normalised relative to CBB staining (top row) or the α -14−3−3 immunoblot (bottom row). **b** Maximum quantum efficiency of PSII ($F_v/F_m$) is compromised in the *tft3 #2-2* mutant upon infection. Tomato leaves were infiltrated with *Pst* DC3000 and $F_v/F_m$ was measured at the indicated times. Markers represent the means and black points represent individual

values ($n = 3$), * indicates significant differences between Prf-SBP-FLAG and *tft3 #2-2* lines using multiple two-sided *T*-tests. **c** Electrolyte leakage following infection with *Pst* DC3000 is compromised in the *tft3 #2-2* mutant. Tomato leaves were infiltrated with *Pst* DC3000 and electrolyte leakage was measured as conductivity (in microSiemens/cm; μS/cm). Markers represent the mean and black points represent individual values ($n = 9$), * indicate significant differences between Prf-SBP-FLAG and *tft3 #2-2* lines using multiple two-sided *T*-tests. **d** The *tft3 #2-2* mutation does not compromise *Pst* DC3000 growth. Bacterial growth of *Pst* DC3000 in leaves of Rio Grande 76 R *prf3*, 76 S, Prf-SBP-FLAG, 76 R and *tft3 #2-2* tomato lines 3 days post infiltration. Error bars represent ±SE ($n = 6$). Significance was calculated using multiple two-sided *T*-tests where * indicates significance as compared to the 76 R line. All experiments were repeated three times and typical results are shown.

control plants (Fig. 4c). We next performed *Pst* DC3000 growth assays for the *tft3 #2-2* mutant line and compared them with results from the susceptible Rio Grande 76 S (which lacks the *Pto* gene) and *prf3* lines, and the resistant Rio Grande 76 R (containing the native Prf/Pto proteins) and the control *prf3/Prf-SBP-FLAG* tomato lines. As expected, we detected higher bacterial growth in the susceptible lines (76 S and *prf3*) compared to the resistant lines (76 R and *prf3/Prf-SBP-FLAG*) (Fig. 4d). Surprisingly, the *tft 3#2-2* mutant line supported the same bacterial titre as 76 R and *prf3/Prf-SBP-FLAG* (Fig. 4d).

These findings indicate that TFT3 is necessary for the full activation of MAPKs and the subsequent inhibition of PSII, while also contributing to transcriptomic reprogramming and PCD formation downstream of Prf/Pto activation. In contrast, the lack of a functional TFT3 is not sufficient to compromise resistance to *Pst* DC3000 in tomato.

## The Prf/Pto complex dissociates into distinct modules following effector recognition

The Prf/Pto complex consists of at least two Prf molecules that interact with each other through their N-terminal domains (Supplementary Fig. 10) and with different Pto family members[18]. As described above, TFT proteins act as facilitators of Prf/Pto-mediated ETI responses by interacting with Pto and MAPKKα . To understand dynamism of this complex, we performed blue native PAGE, co-immunoprecipitations and split-luciferase experiments with all of the known components of the Prf/Pto complex following effector recognition. After confirming the expression of the individual components *in planta* (Supplementary Fig. 11a), a high molecular weight band was observed when complexes were separated on blue native PAGE prior to effector recognition. Following co-expression of either AvrPto or AvrPtoB, a Prf-specific lower molecular weight band was observed, suggesting the

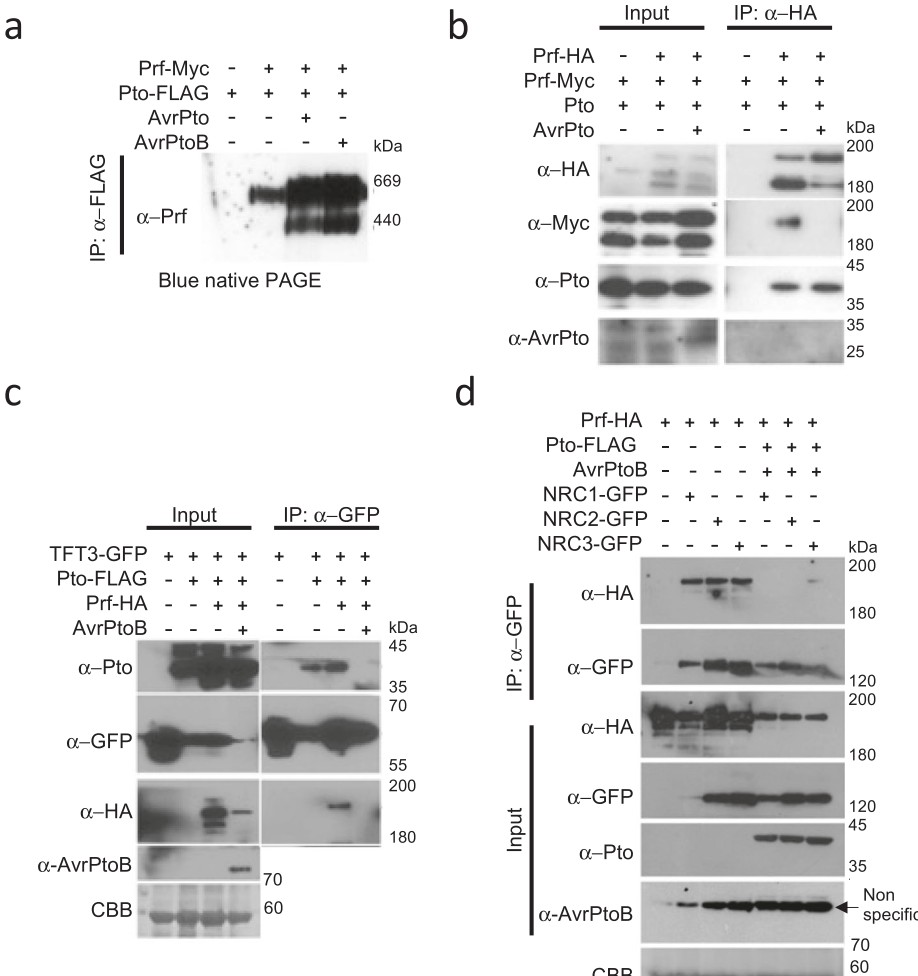

**Fig. 5 | The tomato Prf/Pto complex dissociates following effector recognition.**
**a** Blue native PAGE showing the formation of an additional lower molecular band complex following the expression of *Pseudomonas* effectors AvrPto and AvrPtoB. Prf-Myc and Pto-FLAG constructs were transiently expressed from their native promoters in *N. benthamiana* leaves in the presence or absence of AvrPto or AvrPtoB and proteins were immunoprecipitated (IP) using α-FLAG agarose beads. The eluted fraction was subjected to blue native PAGE. **b** The Prf-Prf dimer dissociates while the Prf-Pto interaction remains intact after effector recognition. Prf-HA, Prf-Myc and Pto constructs were transiently expressed in *N. benthamiana* leaves in the presence or absence of AvrPto and Prf was immunoprecipitated using

α-HA agarose beads. **c** TFT3 and Pto dissociate after effector recognition. TFT3-GFP was co-expressed with Pto-FLAG, Prf-HA and AvrPtoB in *N. benthamiana* leaves and TFT3 was immunoprecipitated using GFP-Trap agarose beads. **d** NRC1,2 and 3 interact with Prf. NRC(1-3)-GFP, Prf-HA, Pto-FLAG and AvrPtoB were transiently expressed in *N. benthamiana* leaves and NRCs-GFP were immunoprecipitated using GFP-Trap agarose beads. Plant leaves were harvested 3 days post infiltration for protein extraction and immunoblots were performed with the antibodies indicated on the left. Coomassie Brilliant Blue (CBB) staining of the membranes was used to verify protein loading. All experiments were repeated three times and typical results are shown.

dissociation of the complex after effector recognition (Fig. 5a). To test this further, we performed co-immunoprecipitations and split-luciferase experiments by expressing Prf and Pto tagged with different epitope tags. In the split-luciferase experiments, a fully functional Prf was reconstituted by expressing the Prf-Nterm and Prf-SCNL fragments[18]. We found that the dimeric interaction between the two Prf molecules was lost when AvrPto or AvrPtoB was co-expressed with Prf (Fig. 5b and Supplementary Fig. 11b). In contrast, the Prf interaction with Pto remained intact in the presence of the effector (Fig. 5b).

Given that we identified TFTs as components of the Prf/Pto complex (Fig. 1), we next investigated the dynamics of TFT3 interaction with Pto in the presence of effectors. For this, we performed co-immunoprecipitations and spit-luciferase experiments between TFT3, Pto and Prf in the presence or absence of the AvrPtoB effector. Consistent with our previous results, in the absence of the effector, Pto, Prf and TFT3 were found as part of the same complex. Following effector recognition, the Prf-Prf interaction (Fig. 5b and Supplementary Fig. 11) and the Pto-TFT3 (Fig. 5c and Supplementary Fig. 12) interaction were

dissociated. In contrast, the Prf-Pto interaction (Fig. 5b) and the TFT3 interaction with the autoactive version of the MAPKKKα (Fig. 2a) remained intact.

It was previously reported that helper NLRs known as NRCs are necessary for cell death formation following recognition of AvrPto and AvrPtoB by the Prf/Pto complex[27]. This prompted us to investigate if NRC1, NRC2 or NRC3 are components of the Prf/Pto complex. We found that Prf interacts with NRC1, NRC2 and NRC3 (Fig. 5d and Supplementary Fig. 13) but interaction of any of the three NRCs was not observed with Pto, TFT3 or MAPKKKα (Supplementary Fig. 14). We further found that the interaction between Prf and NRC1, NRC2 or NRC3 was disrupted once ETI was induced by co-expression of the AvrPtoB effector (Fig. 5d and Supplementary Fig. 15).

These findings illustrate that following effector recognition the Prf/Pto complex dissociates in two signalling modules that contribute to distinct and overlapping aspects of ETI responses (Fig. 6).

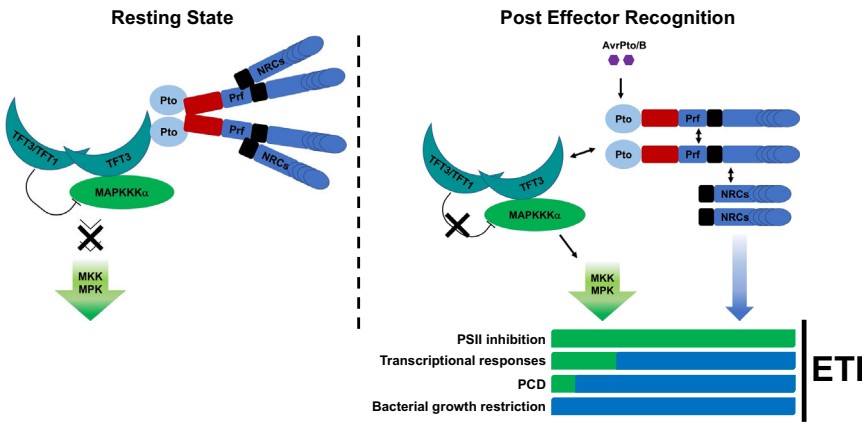

**Fig. 6 | Following effector recognition, the dissociation of the Prf/Pto resistant complex leads to downstream bifurcation of the signalling cascade.** Our model shows that Prf, Pto, TFT1, TFT3, MAPKKKα and NRCs are part of the Prf/Pto resistance complex. Following effector (AvrPto/B) recognition, the complex dissociates into two distinct ETI signalling branches. The TFT3-dependent branch (green bars) is required for full activation of MAPKs, the subsequent inhibition of PSII, the expression of a subset of defence-related genes and partially contributes to programmed cell death (PCD). The NRC-branch (blue bars) also contributes to Prf/Pto-dependent transcriptional responses and is required for full PCD formation and pathogen growth restriction.

## Discussion

Pathogen effectors are recognized by plant NLR proteins to trigger a downstream signalling cascade which includes strong MAP kinase activation, transcriptomic responses and ultimately, localized PCD[32]. However, the precise regulation of downstream signalling is not fully understood. To address this, we exploited the well-studied tomato Prf/Pto-mediated ETI pathway[12] to investigate how NLR activation leads to downstream signalling processes.

In this work, we found that TFT1 and TFT3 interact with Pto, which suggests that they play a role in Prf/Pto-mediated responses. The kinase activity of Pto was not a prerequisite for this interaction but was important for the preferential binding of Pto to TFT3 suggesting that TFT3 rather than TFT1 is the main interactor of Pto (Fig. 1). In contrast, both TFT1 and TFT3 may interact with Pto homologues that have reduced kinase activity and do not contribute to effector recognition[12,15].

The interaction of 14-3-3 proteins with NLR complexes was suggested previously[33] and the Arabidopsis 14-3-3 protein GF14λ has been shown to interact in vitro with the resistance protein RPW8[34]. However, in the case of the Prf/Pto complex, we did not observe an interaction of Prf with any 14-3-3 protein in our experiments, despite the presence of two putative 14-3-3 binding sites in Prf[35,36]. We postulate that the observed dissociation of Pto from TFT3 (Fig. 5) is important for proper downstream phospho-relay signalling which involves activation of MAPKKKα. Our findings are very interesting in the context of Pyrin inflammasome assembly and signalling in humans[37]. Pyrin is an inflammasome protein which is phosphorylated by protein kinase N1/2 (PKN1/2) and autoinhibited by 14-3-3 proteins in a resting state. After infection, bacterial toxins lead to PKN1/2 inactivation and 14-3-3 dissociation, subsequently leading to pyrin inflammasome assembly and activation[38]. The binding of 14-3-3 to phosphorylated Pyrin therefore acts as a molecular switch to turn off Pyrin inflammasome activation. Similarly, prior to activation TFT3 is part of the inactive/repressed state of the Prf/Pto complex and dissociate from the complex after effector recognition (Fig. 5).

Notably, the resolved cryo-EM structure of the animal B-Raf dimer, which is one of the first kinases to be activated in the Ras/Raf/Mek/Erk kinase pathway that transmits extracellular signals to the nucleus, revealed that 14-3-3 proteins can facilitate the inhibition of one kinase whilst maintaining activity of the other[39]. Release of the B-Raf auto-inhibition can lead to disorder in the structure which makes one B-Raf monomer ready for catalysis and further downstream signalling activation. Consistent with this, our data suggest that the

tomato MAPKKKα could also be maintained in an auto-inhibited state by TFT3 while in the Prf/Pto complex. Following effector recognition, MAPKKKα is both activated and dissociated from the Prf/Pto leading to downstream phospho-relay signalling to MAPKs. It is possible that by interacting with both Pto and MAPKKKα, TFT3 serves as an MAPKKKα inhibitor prior to effector recognition while is also necessary for bridging effector recognition by Pto to MAPKKKα activation. Consistent with this model, TFT3 is necessary for the full activation of ETI but the *tft3* mutants do not have any auto-immunity phenotypes (Fig. 4). Thus, by interacting with both Pto and MAPKKKα, TFT3 serves as a bridge connecting effector recognition with subsequent MAPK and ETI activation.

Another 14-3-3 isoform from tomato, TFT7, is known to interact with MAPKKKα and the MAPKK MEKK2 to positively regulate Pto-mediated PCD[19,26]. In this work, we did not detect any interaction between TFT7 and Pto (Fig. 1). Therefore, we reason that TFT3 is important for bridging Pto and MAPKKKα while TFT7 may play a distinct role by bridging MAPKKKα with MEKK2, leading to the intriguing possibility that specific TFT proteins regulate different steps of the Prf/Pto-mediated ETI process. Interestingly, MEKK2 and its downstream MAPK MEK3 appear to interact directly[40], suggesting that it is only the upstream part of the Prf/Pto signalling cascade that is modulated by TFT proteins. Despite the dynamic nature of the TFT3 interaction with the Prf/Pto complex, resistance to *Pst* DC3000 was not compromised in the *tft3* tomato mutant lines (Fig. 4d). Since TFT1 also interacts with Pto, this effect could be due to genetic redundancy. However, Pto preferentially interacts with TFT3 and the *tft3* mutation significantly compromised MAPKs activation and the subsequent PSII inhibition while also reduced PCD formation and regulated part of the transcriptional immune responses of tomato to *Pst* DC3000 (Fig. 4). Our data better fit a model where effector recognition leads to branching of the Prf/Pto-mediated signalling cascade in tomato with each branch contributing specific functions. The TFT3-dependent branch is required for Prf/Pto-mediated activation of MAPKs, the subsequent inhibition of PSII, the expression of a subset of defence-related genes and partially contributes to PCD but is not requited for *Pst* DC3000 growth restriction. The NRC-branch most likely also contributes to Prf/Pto-dependent transcriptional responses and is required for PCD and pathogen growth restriction[27] (Fig. 6).

A similar divergence of signalling has been reported in Arabidopsis after infections with *Pst* DC3000 carrying the effectors, AvrRpm1, AvrRpt2 or AvrRps4 leading to ETI mediated by both MAPK cascades and helper NLRs. In tomato, recognition of the AvrPto and

AvrPtoB effectors by the Prf/Pto complex leads to MAPK activation[41] and inhibition of PSII (Fig. 4b). Similarly, in Arabidopsis, recognition of AvrRPM1 or AvrRpt2 by the RPM1 or RPS2, two CC N-terminal domain NLRs[42,43] leads to MAPKs activation and subsequent inhibition of PSII[29]. Furthermore, recognition of AvrRps4 by RRS1 and RPS4, a pair of Toll/interleukin-1 receptor resistance (TIR) N-terminal NLRs[44], can also lead to MAPK activation and subsequent inhibition of PSII[29]. In tomato following effector recognition, the CC N-terminal domain helper NRCs contribute to transcriptional responses and are required for the full onset of PCD and pathogen growth restriction[27]. In Arabidopsis, the (RPW8)-CC domain containing helper NLRs of the ACTIVATED DISEASE RESISTANCE 1 (ADR1)[9] and N REQUIREMENT GENE 1 (NRG1)[11,45] families contribute to different aspects of the AvrRpm1, AvrRpt2 or AvrRps4-induced ETI in both a redundant and specific manner[46]. Our data show that both MAPKKKα and the helper NRCs are integral parts of the Prf/Pto resistance complex which therefore acts as a hub facilitating signal transduction downstream effector recognition. However, in Arabidopsis it is unclear how the signal is transduced from NLRs to MAPKKKs and helper NLRs. Based on our data, it is tempting to speculate that similar to the Prf/Pto complex, the Arabidopsis NRLs serve as a hub by directly or indirectly interacting with MAPKKKs and helper NLRs.

In summary, our work shows that TFTs and NRCs associate with different components of the Prf/Pto complex prior to its activation and dissociate from the complex following effector recognition. The dynamic nature of these protein-protein interactions together with our tomato phenotypes and transcriptomic data support a model where the cooperation of two parallel pathways stemming from the dissociation of the Prf/Pto complex orchestrate ETI against *Pst* DC3000 in tomato.

## Methods

### Bacteria and plant materials
*Pseudomonas syringae* pv. *tomato* strains and *Agrobacterium tumefaciens* strain GV3101 were grown at 28 °C in King's broth (B) and Luria Broth (LB) medium with appropriate antibiotics, respectively. DCL-suppressed *Nicotiana benthamiana*[47] and tomato Rio Grande *prf3*, *prf3*/Prf-SBP-FLAG, 76 S and 76 R lines were grown in the greenhouse with 16 h of light and average temperature of 24 °C.

### *Agrobacterium*-mediated transient assays in *N. benthamiana*
The tomato *TFT* genes were cloned first cloned into the pDONR-Zeo Gateway entry vector and later shuttled into the pEG104 destination vector using a Gateway LR reaction. The primers used for cloning are listed in Supplementary Table 1. *Pto*, *Prf*, Sl*MAPKKK α* genes and their derivatives were cloned binary vectors pT70 (a pTFS-40 derivative containing the 35 S promoter) with different tags. Each construct was transformed into *A. tumefaciens* strain GV3101 pMp90[23]. Each construct was grown in LB broth under corresponding antibiotic selections at 28 °C overnight. Bacterial cultures were resuspended in infiltration buffer containing 10 mM MES (pH 5.6) and 10 mM MgCl₂ at $OD_{600}$ nm = 0.8. After 48 h, the syringe infiltrated leaves were collected for co-IP experiments.

### Coimmunoprecipitation, immunoblotting and blue native PAGE
Constructs were co-expressed in *N. benthamiana* leaves by *Agrobacterium*-mediated transient expression (as above). After 2–3 days, plant proteins were extracted from harvested leaves in 5 ml of extraction buffer as described earlier[48]. Proteins were incubated with agarose or sepharose beads of indicated epitope tags for 2 h at 4 °C. For GFP and Myc IPs, GFP-Trap and Myc-Trap agarose beads (Chromotek) were used and for FLAG and HA IPs, anti-FLAG M2 agarose and anti-HA agarose (Sigma) beads were used. After washing with washing buffer five times, beads were resuspended with 50 µL of 1 × SDS loading buffer. Proteins were separated on 10% SDS-PAGE gel, transferred onto a PVDF membrane using a semi-dry electroblotter (Bio-Rad), and

detected with antibodies. Mouse anti-FLAG (1:5000, Sigma), anti-cMyc-HRP (1:10000, Santa Cruz), anti-HA-HRP (1:1500, Santa Cruz), anti-GFP-HRP (1:10000, Santa Cruz), rabbit anti-phospho-p44/42 MAPKs (1:5000, Cell Signaling Technology) and rabbit anti-14-3-3 (1:2000, Argisera) were used. The chemiluminescent signal was detected using Amersham ECL-Plus blotting detection system (GE Healthcare). For Blue native PAGE, proteins were extracted in the same way, separated on an Invitrogen 4-16% gradient NativePAGE Bis-Tris Gel, and transferred onto a PVDF membrane following the manufacturer's instructions.

### Split-luciferase assays
For the split luciferase assays, constructs were PCR-amplified from the pT70 vectors with primers containing the BP reaction sites and were subsequently cloned into the pDONRzeo entry vector. *Prf-Nterm* was synthesised containing the BP reaction sites and was cloned into the pDONRzeo vector. All genes were cloned into the pDEST-NLUC[GW] and pDEST-CLUC[GW] binary destination vectors[49] using a Gateway LR reaction. Constructs were transformed into *A. tumefaciens* GV3101 pMp90 and grown in LB broth under corresponding antibiotic selections at 28 °C for 48 h. Cultures were resuspended in infiltration buffer at $OD_{600}$ nm = 0.8, combined in equal ratios to create each combination tested, and infiltrated into *N. benthamiana* leaves using a needleless syringe. Leaves were collected 3 days post-infiltration and luciferase was detected using a Photek HRPCS single photon counting system (Photek Ltd). Luminescence images were analysed using the Image32 software.

### MAPK activation assay
Proteins from pulverised tissue was isolated using extraction buffer (50 mM Tris-HCl, pH 7.5, 100 mM NaCl, 15 mM EGTA, 10 mM MgCl₂, 1 mM Na₂MnO₄·2H₂O, 0.5 mM NaVO₃, 1 mM NaF, 30 mM β-glycerol phosphate, 0.5 mM phenylmethylsulfonyl fluoride, and plant protease inhibitor cocktail [Serva GmbH]). After centrifugation at 30,000 *g* for 30 min at 4 °C, the protein concentration of the supernatant was determined using a Bradford assay. Thirty micrograms of protein were separated on a 10% polyacrylamide gel. Immunoblot analysis was performed using anti-phospho-p44/42 MAPKs (1:5000; Cell Signaling Technology) and peroxidase-conjugated goat anti-rabbit antibody (1:10000; Sigma).

### Mass spectrometry
Immunoprecipitated samples were prepared for mass spectrometry by performing on-beads trypsin digestion. The proteins were reduced with dithiothreitol and alkylated with iodoacetamide. The proteins were subsequently digested with trypsin (Promega) at 37 °C overnight. Subsequently, the peptides were subjected to liquid chromatography-MS/MS analysis. Reverse-phase chromatography was used to separate tryptic peptides prior to mass spectrometric analysis using an Acclaim PepMap µ-precolumn cartridge (300 µm i.d. ×5 mm, 5 µm, 100 Å) and an Acclaim PepMap RSLC (75 µm × 25 cm, 2 µm, 100 Å; Thermo Scientific). Peptides were eluted directly via a Triversa Nanomate nanospray source (Advion Biosciences) into a Thermo Orbitrap Fusion mass spectrometer (Q-OT-qIT; Thermo Scientific). The raw data were processed using MSConvert in the ProteoWizard Toolkit (version 3.0.5759). MS/MS spectra were searched with Mascot engine (Matrix Science; version 2.4.1) against the *N. benthamiana* database (available on request from Sophien Kamoun at the Sainsbury Laboratory; http://www.tsl.ac.uk/staff/sophien-kamoun/) and the common Repository of Adventitious Proteins Database (http://www.thegpm.org/cRAP/index.html). Peptides were generated from a tryptic digestion with up to two missed cleavages, carbamidomethylation of Cys as fixed modifications, and oxidation of Met as variable modifications. Precursor mass tolerance was 5 ppm, and product ions were searched at 0.8-D tolerance[50]. Scaffold (version Scaffold_4.3.2; Proteome Software) was

used to validate MS/MS-based peptide and protein identification. Peptide identifications were accepted if they could be established at greater than 95% probability by the Scaffold Local FDR algorithm.

## ProQ diamond staining of phosphoproteins

Staining of phosphorylated proteins was performed using Pro-Q™ Diamond Phosphoprotein Gel Stain (Invitrogen) following protein separation on SDS-PAGE. The indicated proteins were incubated for 30 min at 30 °C in the kinase buffer containing 25 mM Tris-Cl (pH 7.5), 10 mM $MgCl_2$, 1 mM DTT, 1 mM PMSF, 25 µM ATP. After separation on 10% SDS-PAGE, the gel was fixed with 50% methanol and 10% acetic acid overnight. The gel was washed with water for 30 min and stained with 3x diluted Pro-Q diamond stain (Invitrogen) in the dark for 2 h. The gel was destained four times for 30 min each with 20% acetonitrile, 50 mM sodium-acetate (pH 4.2). The gel was washed again with water for 10 min and was scanned at 400 V using a Typhoon Scanner (GE Healthcare)[51].

## Generation of the CRISPR/Cas9 tomato mutant lines of TFT3

A CRISPR/Cas9 target site for the TFT3 gene was selected using CRISPR-P 2.0 (http://cbi.hzau.edu.cn/CRISPR2/)[52]. To generate the sgRNA construct, sgRNA with a specific target sequence was PCR amplified from pDONR207 (AtU6p-sgRNA with attL1 and attL2; from Jeffrey Dangl, UNC Chapel Hill) using primer sets TFTtarget/sgRNArev. Each PCR product was self-ligated and the product was recombined into pMDC83[53] with Cas9-HA-NLS using LR clonase II (ThermoFisher). The final plasmid was transformed into *A. tumefaciens* strain LBA4404 and then used to transform and regenerate Rio Grande tomato line *prf3*/Prf-SBP-FLAG using standard methods[54]. To confirm the identity of the mutation, DNA surrounding the CRISPR/Cas9 target site was PCR-amplified from the genomic DNA of mutant T1 candidates using gene specific primer set TFTfor/TFTrev and then sequenced (Gene-wiz). The *tft3 2-2* line (*prf3*/Prf-SBP-FLAG/*tft3 2-2*) was characterized in this study.

## Microarray analysis

Leaves of four-week-old tomato *prf3*, *prf3*/Prf-SBP-FLAG and *tft3* 2-2 lines leaves were vacuum infiltrated with *Pst* DC3000 (OD = 0.1) for 6 h; 10 mM $MgCl_2$ infiltration served as control. All infiltrations were carried out in biological triplicate and three technical replicates were used for each. After tissue harvesting and grinding in liquid nitrogen, RNA extraction was performed using a RNeasy kit (QIAGEN) and quality checked using Bioanalyzer analysis. First-strand cDNA synthesis was performed from 2 µg of total RNA using RevertAid reverse transcriptase (Invitrogen). 200 ng of RNA was used for cDNA synthesis and Cy3-labelling using the Low Input Quick Amp Labeling Kit for One-Color Microarray-Based Gene Expression Agilent analysis. 1.65 µg of linearly amplified and labelled cDNA was hybridized for 17 h at 65 °C on 4 × 180k format 60-mer oligonucleotide probes designed against the *S. lycopersicum* cv. Heinz 1706 build 2.4 (annotation 2.5) genome (Agilent design ID = 069672; GEO record GPL21602). Each array contained ~5 probes for 34,619 transcripts. Arrays were imaged using an MS200 microarray scanner with only the 480 nm laser and using the autogain feature of the NimbleScan software. Image (tiff) files were imported into the Agilent Feature Extraction software for quality control assessment, grid alignment and expression value extraction at the probe and transcript level with the RMA algorithm[55] used to carry out background subtraction, quantile normalization and summarization via median polish, and output $log_2$ normalized gene expression levels (see raw and normalised data at GEO (accession number GSE167378). Linear Models for Microarray Data (package *limma* in R[56]) was then used to fit linear models to pairs of samples, identifying genes that contrasted the most between the experimental pairs. Transcripts were termed to be differentially expressed if they showed a Benjamini-Hochberg adjusted *P*-value ≤ 0.05 in the comparison between treatment and control.

## qRT-PCR

cDNA for qPCR analysis was amplified using SYBR FAST (Sigma) and qPCR reactions run in the Realtime PCR System (Agilent Technologies). The transcript level for each gene was standardized based on cDNA amplification of *tubulin* as a reference; primer sequences are in Supplementary Table 1.

## Pathogen assays

Fresh *Pseudomonas syringae* cultures were grown overnight and resuspended to an absorbance at 600 nm of 0.02 in 10 mM $MgCl_2$ and 0.01% Silwet L-77. Four-week-old plants were vacuum-infiltrated with the bacterial culture. To measure bacterial count, leaf discs (10 mm in diameter) were harvested at each time point, samples were homogenized, and serial dilutions were plated on KingB plates containing appropriate antibiotics.

## Electrolyte leakage assays

For the electrolyte leakage assay, four-week-old tomato plants were infiltrated with *P. syringae* DC3000 at an $OD_{600}$ nm = 0.1 with a needleless syringe. Three leaf disks per sample were harvested using a 1 $cm^2$ cork borer three days post-infiltration, for a total of 9 samples per genotype. Leaf disks were washed in distilled water, randomly placed in 12 well plates with 2 ml distilled water and incubated inside a growth chamber. Sample conductivity (in microSiemens/cm) was measured every hour for 8 h with a LAQUAtwin EC-11 (HORIBA Ltd., Kyoto, Japan). Experiments were performed 4 times and data were analysed using a Student's *T*-test.

## Chlorophyll fluorescence imaging

For the Photosystem II chlorophyll fluorescence imaging of tomato leaves a CF Imager (Technologica Ltd, Colchester, UK) was used. Leaves of four-week-old plants were infiltrated with *P. syringae* DC3000 at an $OD_{600}$ nm = 0.1 with a needleless syringe. Plants were returned to the growth chamber for 30 min post-infiltration, leaves were harvested and placed on a square petri dish in the CF Imager. Leaves were dark adapted for 20 min, minimal fluorescence with fully oxidized PSII centres ($F_o$) was measured, followed by a saturating light pulse (6349 µmol m$^{-2}$ s$^{-1}$) for 0.8 s for the maximum dark-adapted fluorescence ($F_m$). Actinic light (120 µmol m$^{-2}$ s$^{-1}$) was applied for 15 min, followed by a saturating pulse for the maximum light adapted fluorescence ($F_m$). Actinic light was applied for another 24 min and was followed by 20 mins of dark. The same 59 min cycle was repeated 15 times and the maximum dark-adapted quantum efficiency of the Photosystem II $F_v/F_m$ was calculated as ($F_m$-$F_o$/$F_m$). Experiments were performed 4 times and data were analysed using a Student's *T*-test

## Gene IDs

DNA sequence information of the genes used in this study can be found in the GenBank/EMBL databases under the following accession numbers: Prf (AAF76308), Pto (AAF76306), SlMAPKKKα (AY500155), SlMAPKKKε (GU192457), TFT1(X95900), TFT2 (X95901), TFT3 (X95902), TFT4 (AJ504807), TFT5 (X95903), TFT6 (X95904), TFT7 (X95905), TFT8 (X98864), TFT9 (X98865), TFT10 (X98866), GRF1 (L09112), GRF2 (M96855), GRF3 (L09110), GRF4 (L09111), GRF5 (L09109), GRF6 (U68545), GRF7 (U60445), GRF8 (U36447), GRF9 (U60444), GRF10 (U36446), GRF11 (AF323920), NRC1 (NP_001234202), NRC2 (XP_004248798.1), NRC3 (XP_004238948.1).

## Reporting summary

Further information on research design is available in the Nature Portfolio Reporting Summary linked to this article.

# Data availability

The microarray data were deposited into the NCBI database (GEO accession: GSE167378). The mass spectrometry proteomics data have

been deposited to the ProteomeXchange Consortium via the PRIDE partner repository with the dataset identifier PXD041469. Source data are provided with this paper.

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

## Acknowledgements

We thank Professor Sophien Kamoun for critically reading the manuscript and for providing materials. We also thank all members of the Ntoukakis' laboratory for fruitful discussions and helpful comments. This research was funded by the Biotechnology and Biological Science Research Council grants (BB/L019345/1) awarded to V.N. and BB/V01627X/1 awarded to M.R.G. and V.N. I.Z, A.J.P. and O.M.N. were funded by the University of Warwick through the Biotechnology and Biological Sciences Research Council, Midlands Integrative Biosciences Training Partnership (BB/M01116X/1). N.U. was funded by the University of Warwick through the Chancellors' International Scholarship scheme. A.D.-F. was supported by the BBSRC/EPSRC funded Warwick Integrative Synthetic Biology Centre (BB/M017982/1) awarded to V.N. G.S. was supported by United States-Israel Binational Science Foundation Grants 2011069 and 2015062. M.L.G. was supported by BBSRC grants BB/H109502/1 and BB/P002145/1. M.B.M was supported by United States-Israel Binational Science Foundation Grant 2015062 and National Science Foundation Grant IOS-2026368. V. N. is also supported by the Royal Society.

## Author contributions

A.H.S., I.Z., G.S., A.M.E.J., M.R.G., M.L.G., M.B.M., J.P.R. and V.N. conceived and designed the experiments. A.H.S., I.Z., A.J.P., A.D-F., D.J.S., J.-G.K., A.B., J.R.G., B.F.C., N.U., O.M.N., A.M.G. and C.-H.W. performed the experiments. A.H.S., I.Z., M.B.M., J.P.R. and V.N. wrote the manuscript. All authors discussed the results and commented on the manuscript.

## Competing interests

The authors declare no competing interests.
