## [Peer Review File · Nature Communications]

Dynamic changes of the Prf/Pto tomato resistance complex following effector recognition.Reviewer #1 (Remarks to the Author):

In this manuscript, the authors describe several aspects of Prf-Pto signaling. Overall, the work seems of high quality but there are several concerning points in the study. The study focuses on the function of scaffolding proteins (TFTs) in regulating Pfr ETI. However, there is little defense-related phenotype in the examined tomato tft3 mutant. The loss of function mutation has no impact on bacterial growth. The transcriptomic comparison shows little impact of TFT3 mutant on ETI induced genes, considering the large set of gene misregulated. TFT3-dependent ETI genes are enriched in catabolism GO and not defense-related which does not argue toward TFT3 having a major role in Prf ETI. The authors show a lower MAPK activation at 6 hours post infection, which is interesting but in the absence of plant phenotype, it is difficult to conclude that TFT3 regulates the MAPK branch of immunity during Prf ETI. Phenotypes in Arabidopsis / Pst avrRpm1 are not really relevant to Pfr/Pto biology, unless authors want to extend 14-3-3 regulation of NLR function to multiple species / NLR. In that case, strong proof in tomato should be presented before extending the concept.

In the transcriptomic experiment, authors claim that genes misregulated in the prf mutant are ETI genes. Strictly, Prf ETI genes should be obtained by comparing infection with DC3000 to infection with DC3000 delta avrPto avrPtoB in WT and prf plants. Otherwise, the noise from such experiment is very strong. The experiment presented here is not bad but it should be used with caution to make claims, which should be backed up by further experiments. The authors claim that genes misregulated only in prf3 or tft3 are effector triggered susceptibility genes. Authors show 559 Prf-dependent ETS genes, 1563 TFT3-dependent ETS genes and 306 genes in the overlap. Following this logic, tft3 mutant are much more impacted by DC3000 and should be more susceptible than prf? 32% of Arabidopsis genes are misregulated in this experiment, the impact of noise and functions that TFT3 may have outside of defense should not be under-evaluated in this part.

Another important relates to the coherence of the study presented here. It seems like the part of the study on protein interaction / phosphorylation and the part that focuses on ETI are not well integrated and could be investigated much further.

- How does phosphorylation and protein interaction relates to defense phenotype? The authors propose that TFTs regulate negatively and positively the MAPKKK activity but the mechanisms studied here, phosphorylation or protein interaction, do not explain this phenomenon. TFT3 requirement is based only on the p44/42 western blot showing reduced MAPK activity at 6 hpi. The authors need more data to make this claim.

- Transcriptomic data could be used much further, what are the genes misregulated in TFT3? Is it linked to MAPK or NLR functions?

- Are the author suggesting that MAPKKKalpha is triggering cell death during Prf ETI (figure 2a and S3), or NRCs as suggested by protein interaction data in figure 5? Presenting this data without further evaluating the impact of MAPKKK / TFTs / NRCs on cell death during infection could be confusing.

- The authors demonstrate that TFT1 and TFT3 interacts with Pto and MAPKKKalpha and postulate that TFT1 and 3 regulate the activity of Prf/Pto and MAPKs during immunity. The author should study the phenotype of the single mutant tft1 , tft3 and the double mutant tft1 / tft3.

- The authors argue that TFT3 act similarly to NRG1 in Arabidopsis, ie focused on cell death and not defense "(NRG1) complex contributes to cell death but is dispensable for pathogen growth restriction". This statement is not in accordance with the most recent research, NRG1 does contribute to defense (see Zhongshou Wu et al 2019 and Svenja Saile et al 2020). This claim should anyway be comforted by additional experiments investigating TFTs involvement in cell death.

Overall, this study is potentially interesting and the work presented is of high quality but the data does not provide a strong requirement for TFTs in defense in tomato, or a mechanism that could explain the regulation of MAPKKK or Prf by TFTs.

Reviewer #2 (Remarks to the Author):

review for Nature comm.

Dynamic changes of the Prf/Pto tomato resistance complex following effector recognition.

Sheikh et al., (Vardis Ntoukakis)

In this manuscript the authors study the events that may lead to downstream signalling upon effector-activation of the tomato NLR Prf, which is guarding and interacting with the kinase Pto. The authors present data that indicate the association of the Pto/Prf complex with (two) 14-3-3 proteins, specifically with TFT3. Further, do the authors show, by co-ip analysis, that TFT3 is associating with the MAPKKKa – an essential component of the Prf-triggered ETI responses. Prf-mediated immunity was shown to be dependent on the presence of the solanaceous specific NRC helper NLRs, but it was not known whether they could associate with Prf/Pto complexes. The authors present co-IP data showing the interaction of NRC1, 2 and 3 with Prf in the pre-activated state and a dissociation of NRCs from Prf upon effector-recognition, when Pto Prf and NRCs are co-expressed. Blue native PAGE analysis by the authors show that Prf and Pto exist in a higher-order molecular weight complex before effector recognition and that upon effector recognition this complex may dissociate to form an additional lower molecular weight complex. Additionally, the authors provide a transcriptomic analysis indicating that the TFT3 protein is required for some, but not all, Prf-triggered transcriptional reprogramming during ETI. However, knockout of TFT3 in tomato did not affect Prf-mediated resistance to infections with Pto DC 3000.

The manuscript is very well written and the data is presented in a clear and understandable way – it was fun reading it.

Given that it is still unclear how NLR activation is translated or redirected into ETI responses and specifically in regard of NLRs that do require the presence of helper NLRs this manuscript touches on a highly important area of plant immunity and the presented idea and results are indeed of interest to a wider range of plant molecular biologist not only in the field of plant immunity. The results are novel and especially timely, given the current new and ground-breaking findings in NLR biology.

The material and methods section is sufficient and clear to reproduce the presented experiments.

However, this reviewer thinks that the currently presented data in the manuscript does not full support the claims made in the title and the abstract. I have outlined my major concerns below:

major issues:

It would be great if authors could provide another method to show the dissociation of the complex members upon effector recognition, for example by FRET-FLIM analysis. Authors should show that the Prf - NRCs interaction is indeed disrupted by effector recognition and not actually by coexpression of Pto already.

It is difficult to conclude anything about complex disruption by looking at and combining IPs done in different infiltrations and not with all components present of the complex. I agree that it might be very difficult to have a sufficient expression of all components, but at least it should be tried.

Further, it would be interesting to test whether loss of TFT3 affects the cell death response rather than the resistance response. Did the authors observe any loss of cell death in their infection analysis done with the tft3 #2-2 line?

Maybe the author could also try to use proximity labelling to see complex changes pre- and post-effector expression/recognition.

The authors should introduce the major classes of NLRs and explain the concept of helper NLRs. I think this is important for their story, since Prf function depends on the helper NLRs of the NRC family.

minor issues:

line 75: it should be 'The N-terminal domain'.

line 136: The experiment referred to here is a co-ip of transiently expressed proteins and therefore the phrasing 'no direct interaction was observed' is not correct here. Co-IP does not test for direct interactions, but rather for things that can associate with each

other in any kind of complex that 'survives' the extraction conditions. Thus, MAPKKKs are not identified in a complex with Pto under the used conditions. Please reword here.

line 223: ..."indicating that they are required..."

line 234: "...ETI transcriptomic responses..."

Suppl Figure 10: please indicate the corresponding band for AvrPtoB. It is not very obvious which one it should be.

Fig. 5b: What are the multiple bands in the Prf blots? Differently phosphorylated versions or splice variants? Only the lower band is the one co-precipitated, right?

Fig. 5c: please indicate the molecular mass (kDa) for all blots shown.

line 278: authors should indicate that RPW8 is not a NLR protein, but rather a resistance protein that shares homology with the N-terminal domain of a subclass of NLRs.

line 279/280: Can the authors provide any reference for the claim that Prf has two potential 14-3-3 binding sites?

line 283 end: ...Pyrin is a inflammasome sensor...

line 297ff: If TFT3 is inhibiting MAPKKKa should a tft3 mutant not have an enhanced MAPK signalling phenotype? Further, if the inhibitory effect of TFT3 on MAPKKKa is only seen in complex with Pfr/Pto one could test if autophosphorylation and transphosphorylation by MAPKKKa is inhibited if all components are mixed together in vitro, right? And if this is Prf independent this should have been seen in suppl figure 6a, correct?

line 300ff: This could also be easily tested in an in vitro pull down assay, since the authors have access to recombinant proteins to test this. or by co-ip of transiently expressed proteins, similar as done for the interaction of Prf-Pto and TFT3 in figure 5c.

line 319: authors should also cite Saile et al., 2020 PlosBiol to support the claim made here.

Can the author elaborate a bit more on why they observed the function of 14-3-3 differs in tomato vs. Arabidopsis or Pfr vs RPM1 mediated ETI?

We thank the Editor and the Reviewers for their comments which helped us to improve the manuscript. In the revised manuscript we have added a significant amount of additional data to support our claims and we have edited the text and figures according to the editor's and reviewer's recommendations.

As requested by the reviewers, we verified all the protein-protein interactions and dissociations using split-luciferase assays. To further investigate the role of TFT3 in Prf/Pto-mediated tomato immunity we performed a time course assays of photosystem II activity and electrolyte leakage upon infection with *Pst* DC3000. We choose these two assays instead of immunoblots of MAPK activation because of their quantitative nature that allows to accurately illustrate the bifurcation of the Prf/Pto signalling cascade. We are pleased to provide these revisions to address the concerns of the reviewers.

Specific responses to reviewer comments **in red**:

Reviewer #1-

The study focuses on the function of scaffolding proteins (TFTs) in regulating Prf ETI. However, there is little defense-related phenotype in the examined tomato tft3 mutant. The loss of function mutation has no impact on bacterial growth.

We thank reviewer 1 for this important suggestion. To address the concerns of the reviewer we have performed time course assays of photosystem II activity and electrolyte leakage assays following infections with *Pst* DC3000. It is well established that activation of MAPKs leads to inhibition of PSII activity, and that electrolyte leakage is good proxy for cell death formation. Our F_v/F_m PSII activity measurements clearly show that the attenuated MAPK activation in the *prf3* and *tft3* #2-2 mutant lines resulted in reduced inhibition of PSII compared to the *prf3/Prf-SBP-FLAG* control lines (Fig. 4b). Furthermore, the electrolyte leakage assays show that the *tft3* #2-2 has reduced cell death at 6 hours post inoculation with *Pst* DC3000 in comparison with the control *prf3/Prf-SBP-FLAG* control lines (Fig 4c). Therefore, TFT3 contributes to Prf/Pto mediated ETI. These results are also summarised in Fig 6.

The transcriptomic comparison shows little impact of TFT3 mutant on ETI induced genes, considering the large set of gene misregulated. TFT3-dependent ETI genes are enriched in catabolism GO and not defense-related which does not argue toward TFT3 having a major role in Prf ETI.

Our transcriptomic experiments show that 26% (775 genes out of 2958) of the *Prf*-dependent transcriptional responses to *Pst* DC3000 are *TFT3*-dependent. We believe this is a significant percentage and we have highlighted and explained this in the text. Although *TFT3*-dependent ETI genes are enriched in catabolism GO terms, tomato GO terms are less well-defined than Arabidopsis, and therefore this analysis is somewhat limited. Nevertheless, these genes are clearly part of the *Prf*-dependent tomato transcriptional response to *Pst* DC3000 infection.

The authors show a lower MAPK activation at 6 hours post infection, which is interesting but in the absence of plant phenotype, it is difficult to conclude that TFT3 regulates the MAPK branch of immunity during Prf ETI

We believe that collectively our data support our amendment statement:

"These findings indicate that TFT3 is necessary for the full activation of MAPKs and the subsequent PSII inhibition, while also contributing to transcriptomic reprogramming and

PCD formation downstream of Prf/Pto activation. In contrast, the lack of a functional TFT3 is not sufficient to compromise resistance to *Pst* DC3000 in tomato.” (Line 232)

Phenotypes in Arabidopsis / Pst avrRpm1 are not really relevant to Pfr/Pto biology, unless authors want to extend 14-3-3 regulation of NLR function to multiple species / NLR. In that case, strong proof in tomato should be presented before extending the concept.

We agree with the reviewer and so to avoid confusion we have removed the Arabidopsis data from the manuscripts.

In the transcriptomic experiment, authors claim that genes misregulated in the prf mutant are ETI genes. Strictly, Prf ETI genes should be obtained by comparing infection with DC3000 to infection with DC3000 delta avrPto avrPtoB in WT and prf plants. Otherwise, the noise from such experiment is very strong. The experiment presented here is not bad but it should be used with caution to make claims, which should be backed up by further experiments.

We can see the point of the reviewer, but the suggested experiment comes with a different set of problems. The *Pst* DC3000 delta avrPto avrPtoB has a significantly reduced ability to suppress PTI (He *et al.* 2006) and we now know that PTI contributes to the robustness of ETI (Ngou *et al.* 2021). Furthermore, we have previously shown that the Prf complex in addition to AvrPto and AvrPtoB can recognise an unknown *Pst* DC3000 effector (Gutierrez *et al.* 2009) which means that infection with *Pst* DC3000 delta avrPto avrPtoB can still activate tomato ETI. Therefore, the comparison of *Pst* DC3000 to infection with *Pst* DC3000 delta avrPto avrPtoB in WT and *prf* plants will be also problematic. To clarify this issue, we have changed the text and the figures to reflect that these are *Prf*-dependent transcriptional responses to *Pst* DC3000 and not *Prf*-dependent ETI responses.

The authors claim that genes misregulated only in prf3 or tft3 are effector triggered susceptibility genes. Authors show 559 Prf-dependent ETS genes, 1563 TFT3-dependent ETS genes and 306 genes in the overlap. Following this logic, tft3 mutant are much more impacted by DC3000 and should be more susceptible than prf? 32% of Arabidopsis genes are misregulated in this experiment, the impact of noise and functions that TFT3 may have outside of defense should not be under-evaluated in this part.

We appreciate the comments of the reviewer. Indeed, it is intriguing that we find 559 *Prf*-exclusively altered genes versus 1563 *TFT3*-exclusively altered genes. However, we believe this is not a direct reflection of the respective proteins' involvement in tomato immunity, as our experiments show (Fig. 4). The increased number of *TFT3*-dependent genes most likely reflects the role of 14-3-3 proteins in PTI and other biological functions such as hormonal regulation. As we explain in the text (line 187) from the 6445 Differentially Expressed Genes in response to *Pst* DC3000 infection in the *prf3/Prf-SBP-FLAG* resistant lines, 3487 were also DEGs in the *prf3* susceptible mutant lines, therefore their expression did not rely on the presence of the *Prf* gene. In contrast, 2958 genes that were DEGs in the *prf3/Prf-SBP-FLAG* lines were not differentially expressed in the *prf3* mutant lines, hence these genes represent the *Prf*-dependent transcriptional responses to *Pst* DC3000. The majority of these genes (2183 of the 2958, 74%) were also DEGs in the *tft3 #2-2* mutant lines, hence their expression is *Prf*-dependent but *TFT3*-independent. In contrast, 775 (26%) genes were differentially expressed only if both *Prf* and *TFT3* genes were present, hence their expression is *Prf*- and *TFT3*-dependent.

We agree with the reviewer that to confidently assign these 1563 genes to ETS, we would need to test *Pst* DC3000 vs. *Pst* DC3000 delta *avrPto avrPtoB* in WT and *prf* plants. But as we stated above this experiment also comes with some significant caveats. To avoid confusion, we have relabelled these 1563 genes as “TFT3-dependent genes not involved in *Prf*-dependent transcriptional responses to *Pst* DC3000”, whereas the 569 *Prf*-dependent genes are not involved in *Prf*-dependent transcriptional responses to *Pst* DC3000.

Finally, to clarify, this assay was performed in tomato plants, with ~35,000 annotated genes (PMID: 22660326).

How does phosphorylation and protein interaction relate to the defense phenotype? The authors propose that TFTs regulate negatively and positively the MAPKKK activity but the mechanisms studied here, phosphorylation or protein interaction, do not explain this phenomenon. TFT3 requirement is based only on the p44/42 western blot showing reduced MAPK activity at 6 hpi. The authors need more data to make this claim.

We have previously shown how protein phosphorylation and interactions relate to phenotypes for *Prf*-*Prf* interactions and *Prf*-*Pto* interactions (Ntoukakis *et al.* 2009,2013 and Gutierrez *et al.* 2009). In the revised Fig. 4 we now show the role of TFT3 in *Prf*-mediated tomato immunity in more detail.

Transcriptomic data could be used much further, what are the genes misregulated in TFT3? Is it linked to MAPK or NLR functions?

We appreciate the advice of the reviewer. We have now added Gene Ontology (GO) terms analysis of *Prf*-dependent TFT3-independent genes (2183) as well as *Prf* and TFT3-dependent genes (775) (see Supplementary Fig. 9). We found that TFT3-dependent genes were enriched in GO terms associated with catabolic processes. As discussed earlier, tomato GO terms are less well-defined than *Arabidopsis* GO terms, and therefore this analysis is somewhat limited, however, it still reinforces the idea that *Prf* response branches out in two distinct pathways, one assisted by TFT3 and one TFT3 independent.

*Are the author suggesting that MAPKKK α is triggering cell death during *Prf* ETI (figure 2a and S3), or NRCs as suggested by protein interaction data in figure 5? Presenting this data without further evaluating the impact of MAPKKK / TFTs / NRCs on cell death during infection could be confusing.*

There is no clear answer here. MAPKKK α N term Δ triggers cells death (Supplementary Fig. 3) while at the same time NRCs are necessary for cell death (Wu *et al.* 2016). Our electrolyte leakage assays show that the *tft3* #2-2 has reduced cell death (Fig. 4c). Overall, the data suggest that cell death is primarily regulated by NRCs but the TFT3/ MAPKKK α also contributes to PCD. We have conveyed this message in the revised Fig. 6.

*The authors demonstrate that TFT1 and TFT3 interacts with *Pto* and MAPKKK α and postulate that TFT1 and 3 regulate the activity of *Prf/Pto* and MAPKs during immunity. The author should study the phenotype of the single mutant *tft1*, *tft3* and the double mutant *tft1 / tft3*.*

We appreciate the reviewer's suggestion, but our experiments have shown that *Pto* preferentially binds to TFT3 over TFT1 (Fig 1b) and that TFT3 specifically homo-dimerises and hetero-dimerises with only TFT1 (Fig. 2c). In contrast, the interaction of TFT1 with other TFTs was less specific as it interacted and formed heterodimers with all tested TFTs (Supplementary Fig. 6). Therefore, we focused our effort on TFT3 for which we now have clear phenotypes (Fig 4). Creating 2 additional tomato mutants would take a substantial

amount of time and effort and is beyond the frame of this work. Most importantly, given the lack of TFT1 specificity we are not convinced that these mutants will be informative.

The authors argue that TFT3 act similarly to NRG1 in Arabidopsis, ie focused on cell death and not defense “(NRG1) complex contributes to cell death but is dispensable for pathogen growth restriction”. This statement is not in accordance with the most recent research, NRG1 does contribute to defense (see Zhongshou Wu et al 2019 and Svenja Saile et al 2020). This claim should anyway be comforted by additional experiments investigating TFTs involvement in cell death.

Based on our new data we have now considerably changed the discussion, comparing AvrPto/B recognition in tomato to AvrRpm1, AvrRpt2 and AvrRps4 recognition in Arabidopsis.

Reviewer #2-

It would be great if authors could provide another method to show the dissociation of the complex members upon effector recognition, for example by FRET-FLIM analysis. Authors should show that the Prf - NRCs interaction is indeed disrupted by effector recognition and not actually by coexpression of Pto already. It is difficult to conclude anything about complex disruption by looking at and combining IPs done in different infiltrations and not with all components present of the complex. I agree that it might be very difficult to have a sufficient expression of all components, but at least it should be tried.

We thank the reviewer for this interesting comment. We have used split-luciferase to validate all protein-protein interactions and dissociations following effector recognition. In these experiments, that are much more sensitive than co-IPs and therefore allow the agroinfiltration of multiple constructs, we included all the components of the complex (see Supplementary Figs. 11, 12, 13 and 15). We want also to highlight to the reviewer that *N. benthamiana* has a partially functional Prf, Pto and fully functional MAPKKK α and 14-3-3 proteins therefore all components of the Prf/Pto complex are present in the Co-IP experiments.

Further, it would be interesting to test whether loss of TFT3 affects the cell death response rather than the resistance response. Did the authors observe any loss of cell death in their infection analysis done with the tft3 #2-2 line?

We thank the reviewer for this suggestion and have included this data in Fig. 4c. Electrolyte leakage assays show that following infections with *Pst* DC3000 the *tft3* #2-2 has reduced cell death in comparison to control *prf3/Prf-SBP-FLAG* lines. At the same time, our F_v/F_m PSII activity measurements clearly show that the attenuated MAPK activation in the *prf3* and *tft3* #2-2 mutant lines results in reduced inhibition of PSII compared to the *prf3/Prf-SBP-FLAG* control lines (Fig. 4b).

Maybe the author could also try to use proximity labelling to see complex changes pre- and post-effector expression/recognition.

This is an interesting suggestion but not necessary for the scope of this manuscript. Our split-luciferase assays together with the Co-IPs and the blue native gel can effectively demonstrate the dynamic change in the Prf/Pto complex composition.

The authors should introduce the major classes of NLRs and explain the concept of helper NLRs. I think this is important for their story, since Prf function depends on the helper NLRs of the NRC family.

Good idea, we have now done that in our discussion.

minor issues:

We thank the reviewer for taking the time to identify minor issues

line 75: it should be 'The N-terminal domain'.

Fixed

line 136: The experiment referred to here is a co-IP of transiently expressed proteins and therefore the phrasing 'no direct interaction was observed' is not correct here. Co-IP does not test for direct interactions, but rather for things that can associate with each other in any kind of complex that 'survives' the extraction conditions. Thus, MAPKKKs are not identified in a complex with Pto under the used conditions. Please reword here.

Fixed

line 223: "...indicating that they are required..."

This part of the manuscript has been removed.

line 234: "...ETI transcriptomic responses..."

Fixed

Suppl Figure 10: please indicate the corresponding band for AvrPtoB. It is not very obvious which one it should be.

Fixed

Fig. 5b: What are the multiple bands in the Prf blots? Differently phosphorylated versions or splice variants? Only the lower band is the one co-precipitated, right?

The multiple bands of Prf are a product of degradation; we have previously published similar immunoblots. It is the upper band that co-IP since the degradation is at the N-term of Prf. This blot was cropped wrongly, we apologise for that.

Fig. 5c: please indicate the molecular mass (kDa) for all blots shown.

Fixed

line 278: authors should indicate that RPW8 is not a NLR protein, but rather a resistance protein that shares homology with the N-terminal domain of a subclass of NLRs.

We have now included this in the discussion.

line 279/280: Can the authors provide any reference for the claim that Prf has two potential 14-3-3 binding sites?

We have added the references into the text.

line 283 end: ...Pyrin is a inflammasome sensor...

Fixed

line 297ff: If TFT3 is inhibiting MAPKKKa should a tft3 mutant not have an enhanced MAPK signalling phenotype? Further, if the inhibitory effect of TFT3 on MAPKKKa is only seen

in complex with Pfr/Pto one could test if autophosphorylation and transphosphorylation by MAPKKKa is inhibited if all components are mixed together in vitro, right? And if this is Prf independent this should have been seen in suppl figure 6a, correct?

This is an interesting point. We believe that TFT3 inhibits MAPKKKa while at the same time MAPKKKa needs to receive an activation signal from the Prf/Pto complex. Therefore, despite removing TFT3 inhibition we do not observe increased MAPKKKa activation because it cannot receive the signal.

line 300ff: This could also be easily tested in a in vitro pull down assay, since the authors have access to recombinant proteins to test this. or by co-ip of transiently expressed proteins, similar as done for the interaction of Prf-Pto and TFT3 in figure 5c.

We have attempted this experiment but due to the very low expression levels of MAPKKK α (see Fig. 2) the result was inconclusive. In general, it is very hard to co-IP proteins that are not directly interacting.

line319: authors should also cite Saile et al., 2020 PlosBiol to support the claim made here.

We have removed this statement from the manuscript, but we have referenced Saile et al., 2020 PlosBiol in our discussion.

Can the author elaborate a bit more on why they observed the function of 14-3-3 differs in tomato vs. Arabidopsis or Pfr vs RPM1 mediated ETI?

We have removed these data and the relevant sections from the manuscript.

Reviewer #1 (Remarks to the Author):

My main issue with the previous version of the paper, the absence of tft3 susceptibility phenotype, has been addressed by the authors. In the current version, the author propose a mechanism of MAPK activation by Prf/Pto/TFT3 which is both timely and important. Thank you for your hard work.

Reviewer #2 (Remarks to the Author):

The authors have responded to all my comments and concerns and included some more analysis, specifically to support their interaction analysis and to further show a potential phenotype of the tft3 mutant. They further have left the Arabidopsis data out to better stream line the manuscript flow. This all is very appreciated.

However, I am still not convinced whether the observed effect and interpretation of the TFT function is really supported by the data presented. The phenotype of the tft3 mutant is only strong in regard to transcriptional changes, but the immune phenotype s are really weak and most strikingly do not translate into an immune phenotype. Is this really all explained by redundancy (see comment below)? This story is still very interesting and the experiment are presented in a solid way. However, I am not sure how relevant this all is.

Anyways, this does not necessarily mean that this manuscript is not of broader interest. It just, in my opinion, does not really advance the field in regard of identifying a new mechanism. Its more like a first stage of something interesting and new, i.e. the interaction of the NRCs with the pre-activation Pto/Prf complex is interesting.

well, please see here my two major concerns:

Fig 1b: I think that here one control is missing. It would be good to show the IP with Pto KD (D164N) and TFT1 only to support the conclusion that the preferential binding is dependent on the kinase activity of Pto. also this result was not discussed in the manuscript. What does it mean for the function of this complex?

line 299ff: Here the authors suggest/hypothesize that TFT1/3 binding to Pto and MAPKKK keeps the Pto/Prf complex and the MAPKKK pathway inactive or repressed and that the effector recognition leads to a dissociation of TFT3/1 and thus to full ETI activation. But would this not also suggest that in a tft3 mutant the Pto/Prf complex would be activated and we would also see a loss of negative regulation of the MAPK pathway and thus a unwanted activation of immunity? Can this just be explained by redundancy of TFT1 and TFT3? I dont think so, because otherwise the effect of the loss of tft3 would - at leats in my opinion - not lead to so observable transcriptional and PSII phenotypes, right? Can the authors comment on this please in their discussion.

We thank the Editor and the Reviewers for their comments which helped us to improve the manuscript. In the revised manuscript we have performed the additional experiment requested by reviewer 2 and we have edited the text according to the editor's and reviewer's 2 recommendations.

As requested by the reviewer 2, we verified the interaction of TFT1 with Pto and PtoD164N (see Sup. Fig 2). As expected, TFT1 interacts with Pto and PtoD164N.

Specific responses to reviewer comments **in red**:

Reviewer #1

My main issue with the previous version of the paper, the absence of *tft3* susceptibility phenotype, has been addressed by the authors. In the current version, the author propose a mechanism of MAPK activation by Prf/Pto/TFT3 which is both timely and important. Thank you for your hard work.

We thank reviewer 1 for appreciating our efforts and the significance of our work.

Reviewer #2

The authors have responded to all my comments and concerns and included some more analysis, specifically to support their interaction analysis and to further show a potential phenotype of the *tft3* mutant. They further have left the Arabidopsis data out to better stream line the manuscript flow. This all is very appreciated.

We thank the reviewer for appreciating our efforts and our willingness to follow his recommendations. It took a significant amount of effort and resources to verify all the protein-protein interactions.

However, I am still not convinced whether the observed effect and interpretation of the TFT function is really supported by the data presented. The phenotype of the *tft3* mutant is only strong in regard to transcriptional changes, but the immune phenotype s are really weak and most strikingly do not translate into an immune phenotype. Is this really all explained by redundancy (see comment below)?

We respectfully disagree with the reviewer. The immunity phenotypes of the *tft3* mutant include strong transcriptional differences (Fig. 3), a clear reduction of MAPKs (Fig. 4a) and a reduced suppression of PSII compared to the *prf3/Prf-220 SBP-FLAG* control line that is identical to the reduced suppression of PSII in the *prf3* mutant lines (Fig. 4b). Therefore, redundancy can potentially explain the small difference in electrolyte leakage between *tft3* mutant and *prf3/Prf-220 SBP-FLAG* control line (Fig. 4c) and the similar growth of *Pst* DC3000 in *tft3* mutant lines and *prf3/Prf-220 SBP-FLAG* control lines (Fig. 4d) but cannot explain the transcriptional differences, the differences in MAPKs activation and the subsequent inability to inhibiting plant photosynthesis. We believe our data better fit a model where effector recognition leads to branching of the Prf/Pto-mediated signalling cascade in tomato with each branch contributing specific functions.

This story is still very interesting and the experiment are presented in a solid way. However, I am not sure how relevant this all is. Anyways, this does not necessarily mean

that this manuscript is not of broader interest. It just, in my opinion, does not really advance the field in regard of identifying a new mechanism. Its more like a first stage of something interesting and new, i.e. the interaction of the NRCs with the pre-activation Pto/Prf complex is interesting.

We thank the reviewer for recognising the quality of our work. Doing biochemistry with NLRs is always a major challenge. We know for some time that MAPKs get activated following effector recognition and more recently helper NLRs have emerged as an important player in ETI. We still do not know if all NLRs use helper NLRs but we do know that many do. Our work shows that both MAPKs and NLRs are part of the immune complex which has not been shown before, this enhances our understanding of ETI signalling.

Furthermore, in addition to NLR networks models where multiple sensor NLRs can share a helper NLR, our results put forward a model where preformed NLRs complexes control ETI. The network model implies that multiple NLRs can share a helper NLR which in terms assumes that small amount of helper NLRs are required for functional immunity. Our data suggest that although a single helper NLR may contribute to signalling from multiple sensors, the amount of the helper NLR participating in each sensor NLR signalling is predetermined. This is important because suggests that helper NLRs may be a bottleneck in the full onset of ETI signalling. A fact that should be considered when we introduce multiple sensor NLRs in plants.

Our data also raise a significant question in ETI signalling. Is there crosstalk between MAPKs and helper NLRs activation? Our results showing that both helper NLRs and MAPKKK α are part of the recognition complex, suggesting that most likely this is the case. We are currently investigating this possibility. This is important because if we can disentangle cell death formation from other immune responses such as suppression of PSII, we can potentially alleviate the developmental trade-offs that follow activation of immunity.

well, please see here my two major concerns:

Fig 1b: I think that here one control is missing. It would be good to show the IP with Pto KD (D164N) and TFT1 only to support the conclusion that the preferential binding is dependent on the kinase activity of Pto. also this result was not discussed in the manuscript. What does it mean for the function of this complex?

To address the concerns of the reviewer we have performed the requested experiment. In supplementary figure 2a we show that TFT1 interacts with Pto and PtoD164N. We have also expanded the discussion of these results in the text.

line 299ff: Here the authors suggest/hypothesize that TFT1/3 binding to Pto and MAPKKK keeps the Pto/Prf complex and the MAPKKK pathway inactive or repressed and that the effector recognition leads to a dissociation of TFT3/1 and thus to full ETI activation. But would this not also suggest that in a tft3 mutant the Pto/Prf complex would be activated and we would also see a loss of negative regulation of the MAPK pathway and thus a unwanted activation of immunity?

We apologise for the lack of clarity. In our model TFT3 plays a dual role where in resting state inhibits autoactivation of MAPKKK α while after effector recognition is necessary for

bridging effector recognition by Pto to MAPKKK α activation. In the revised version of the manuscript, we have fully explained this point in the text

Can this just be explained by redundancy of TFT1 and TFT3? I dont think so, because otherwise the effect of the loss of *tft3* would - at least in my opinion - not lead to so observable transcriptional and PSII phenotypes, right? Can the authors comment on this please in their discussion.

We have expanded our discussion in the text addressing this point. We believe that genetic redundancy is highly unlikely due to Pto kinase-dependent preferential binding to TFT3 and because the *tft3* mutation significantly compromised MAPKs activation and subsequent PSII inhibition while also reduced PCD formation and regulated part of the transcriptional immune responses of tomato to Pst DC3000. As explained above, our data better fit a model where effector recognition leads to branching of the Prf/Pto-mediated signalling cascade in tomato with each branch contributing specific functions.

Reviewer #2 (Remarks to the Author):

The authors have addressed and responded to all my concerns and I appreciate their discussion and argumentation. This manuscript will be of interest to many plant (immune-)biologists and will be forming a basis for further studies regarding the function and interaction of 14-3-3 proteins for immunity and with immune components, respectively.
This reviewer has no further concerns.